# Agentic Framework for Multilingual Summarization of Government Notifications Using Firefly-Optimized Dual Transformers

M. Prasannakumar[1,] R. Mohanraj[2,] S. Andal[3], B. Rajalingam[4] and R. Santhoshkumar[5]

[1,3] Assistant Professor, [2,4] Associate Professor, [5]Professor
[1,4]Department of CSE, School of Computing, Vel Tech Rangarajan Dr. Sagunthala R&D Institute of Science and Technology, Avadi, Chennai, Tamil Nadu-600062
[2]Department of CSE(AI), Sri Venkateshwara College of Engineering & Technology, Chittoor, Andra Pradesh-517 127
[3] Department of CSE, Francis Xavier Engineering College, Vannarapettai, Tirunelveli, Tamil Nadu -627003
[5] Department of CSE, Sree Dattha Group of Institutions, Sheriguda, Ibrahimpatnam, Hyderabad – 501510
[1]iamsidharthprasanna@gmail.com, [2]Mohanraj254@gmail.com ,[3]andal@francisxavier.ac.in,
[4]rajalingam35@gmail.com , [5]santhoshkumar.aucse@gmail.com

**Abstract.** In India, the majority of Government administration announcements and notifications are published in English, making it problematic for low-level and rural employees to comprehend official communications. We recommend a novel framework called Cross-Lingual Government Information Summarization using Dual Transformer Network Optimized with Firefly-Enhanced Grey Wolf Optimization (CLGIS-DTN-FEGWO) to address this issue. The system starts by normalizing domain-specific terms using Contextual Noise Reduction and SentencePiece Tokenization (CNR-ST). To provide high-fidelity multilingual output, the processed text is translated using a Dual Transformer Network (DTN), which combines XLM-R embeddings with Neural Machine Translation. After that, brief abstractive summaries are produced by a Graph-Attentional Summarizer (GAS). Through Firefly-Enhanced Grey Wolf Optimization (FEGWO), the DTN is optimized to improve efficiency and accuracy. According to experimental results, CLGIS-DTN-FEGWO outperforms baselines like mBART-50, XLM-R, BART, and PEGASUS by 4–8%, achieving BLEU = 91.50 and ROUGE scores above 90. The contributions of dual-transformer refinement, graph-attentional summarization, and FEGWO optimization are further validated by ablation studies, guaranteeing excellent, contextually accurate translations and summaries. Explainable AI attention heatmaps increase openness, and secure APIs that protect privacy protect sensitive data. With its multilingual support, offline accessibility, and user-friendly interface, the framework is implemented as a web application that successfully bridges the digital language gap in Indian government communication.

**Keywords:** Cross-lingual access, Dual Transformer Network, Graph-Attentional Summarization, Grey Wolf Optimization, explainable AI;

# 1        Introduction

In India's governance environment, language accessibility continues to be one of the key obstacles. Even though digital platforms have made it possible for government circulars, tenders, and welfare announcements to be distributed more quickly, the fact that these papers are frequently issued just in English limits their applicability to rural employees and residents. Research shows that many Indians have difficulty understanding official English-language content, which lowers their knowledge of government programs and makes public service delivery less inclusive [1], [2]. To fill these shortcomings, recent advances in cross-lingual information access (CLIA) have tried to use neural machine translation (NMT) and automatic summarization. Although summarizing models such as TextRank [6], BART [7], and PEGASUS [8] can provide concise and informative summaries, multilingual environments have demonstrated the utility of transformer-based architectures [3], mBART [4], and XLM-R [5]. Notwithstanding these achievements, there are three primary disadvantages to government communication using the models currently in use:  1. Insufficient handling of domain-specific administrative terminology; 2. No scalability across many Indian languages; and 3. Insufficient attention to explainability and privacy concerns.

To address these issues, our study develops a deep learning-based cross-lingual summarization system for government notifications. Unlike earlier attempts, the system includes advanced preprocessing, optimal translation, abstractive summarization, explainability, privacy, and adaptability techniques. Moreover, a multilingual web application is developed to give end users outputs in easily accessible formats. The remainder of this paper is organized as follows: Conclusions and the future scope are covered in Section V. Experimental evaluation is given in Section IV. The proposed approach is described in Section III. Relevant work is evaluated in Section II.

# 2        Literature Review

The previous research on Indian language cross-lingual information access and summarization reveals both advantages and disadvantages. A statistical machine translation (SMT)-based framework for government documents was proposed by Joshi and Pandey [9], however it had trouble with grammatical consistency and administrative terminology. Datta et al. [11] created MILDSum, a benchmark dataset for summarizing Indian court rulings into Hindi, whereas Urlana et al. [10] presented PMIndiaSum, a multilingual summation corpus spanning 14 Indic languages, to solve resource constraints.

In a similar vein, Aralikatte et al. [12] showed difficulties even for sophisticated abstractive models when they provided Vārta, a sizable dataset for Indic headline generation. For cross-lingual summarizing tasks, Bhatnagar et al. [13] prioritized data quality by automatically retrieving and filtering data. For multilingual and abstractive summarization, transformer-based pre-trained models like mBART [14], BART [15], and PEGASUS [16] have demonstrated excellent performance.  However, domain-specific contexts, official government notices, and Indian languages with limited resources all lessen their efficacy. Together, these efforts lay the groundwork for multilingual

summarization, but they also highlight enduring difficulties with supporting low-resource languages, handling domain-specific terminology, and guaranteeing contextual accuracy. These issues are addressed by the approach we have suggested.

**Table 1.**    Literature Survey

| Work | Focus / Dataset | Approach / Model | Key Limitations |
|---|---|---|---|
| **Joshi & Pandey [9]** | Government documents (English → Indic) | SMT-based translation | Poor Grammar; Weak domain handling |
| **Urlana et al. [10]** | PMIndiaSum (14 Indic language) | Multilingual Corpus | Limited Abstractive Support |
| **Datta et al. [11]** | MILDSum (Legal judgments) | Cross-lingual Summarization | Restricted to Legal Domain |
| **Aralikatte et al. [12]** | Vārta (41M pairs) | Headline Generation | Challenging for SOTA Models |
| **Bhatnagar et al. [13]** | Data Retrieval & Filtering | Quality enhancement | Focused on Preprocessing |
| **Liu et al. [14]** | mBART | Pretrained Transformer | High cost; weak on low-resource language |
| **Lewis et al. [15]** | BART | Abstractive model | Needs domain adaptation |
| **Zhang et al. [16]** | PEGASUS | Gap-sentence pretrain | Poor for low-resource Indic language |

## 3    Proposed Method

The goal of the proposed CLGIS-DTN-FEGWO system is to ensure accuracy, explainability, and privacy while automatically translating and summarizing government notifications into several Indian languages. Feature extraction, dual transformer translation, graph-attentional summarization, data preprocessing, and FEGWO-based hyperparameter optimization are the five main steps of the methodology, which is then deployed in a web application. The following diagram Fig 1 explain the overall architecture of the CLGIS-DTN-FEGWO system.

### 3.1    Data Preprocessing

Let the input document be $D = \{ d_1 , d_2, \ldots . d_n \}$ , where $d_i$ represents a sentence. The Preprocessing includes,

1.   Tokenization: Split sentences into tokens:

$$Term_i = Tokenize\ (d_i) \qquad (1)$$

2.   Stop-word removal and normalization:

$$N_i = Normalize\ (Term_{i\ )} \qquad (2)$$

The above Equation 2 standardizes and eliminates noise from the text for improved summarization and translation.

3.   POS tagging and the extraction of domain-specific terminology:

$$P_i\ = \{P_1\ , P_2\ ,\ ...., P_m\ \}\ \text{for each}\ N_i \qquad (3)$$

In order to maintain meaning in translation, the aforementioned equation 3.3 extracts important administrative terms.

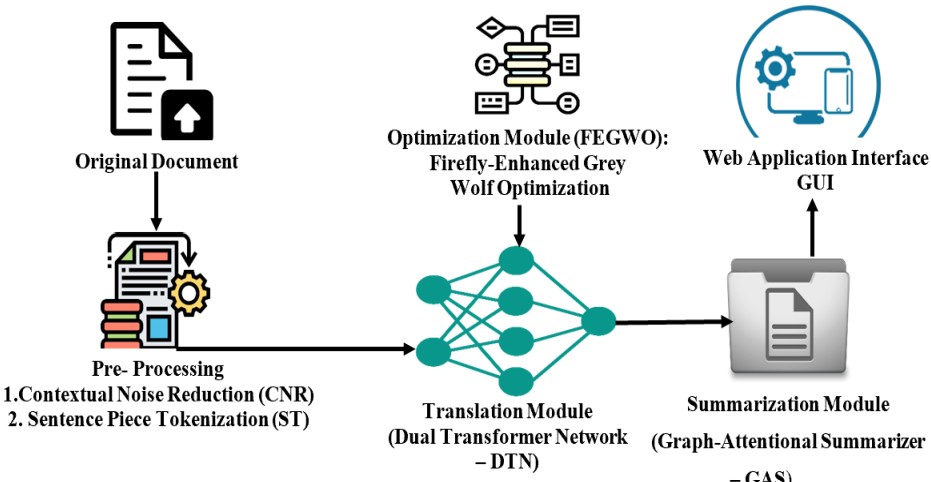

**Fig. 1.** Overall System Architecture

## 3.2    Feature Extraction

The following Equation 4   used to generate word embeddings in XLM-R for multi-lingual contexts, and mean pooling is used to obtain the sentence-level embeddings. Every sentence's semantic representation in the multilingual space is captured by the following equation 5.

$$FE_i = XLM - R\ (\ N_i\ )\ \mathcal{E}\ R^d \qquad (4)$$

$$Sentence_i = \frac{1}{|N_i\ |}\ \sum_{j=1}^{N_i} E_{ij} \qquad (5)$$

### 3.3 Dual-Transformer Translation

The proposed system uses a dual-transformer architecture. The first transformer for source-to-target translation $(Trans_{S \to t})$ and one for target refinement $(Target_{ref})$:

$$Y_t = Trans_{S \to t} (S_i) \qquad (6)$$

$$\widehat{Y_t} = Target_{ref} (Y_t, S_i) \qquad (7)$$

A base translation is produced by the first transformer, and domain-specific terminology and fluency are improved by the second. The above equation 6 , 7 used for the translation.

### 3.4 Graph-Attentional Summarization

Let the translated sentences $\widehat{Y_t} = (\widehat{y_1}, \widehat{Y_2}, \dots \widehat{y_m})$ form a graph G = (V, E), where each sentence is a node and edges represent semantic similarity:

$$Semantic\_weight_{ij} = \cos(\widehat{y_i} \, \widehat{y_j}) \qquad (8)$$

The graph attention network (GAT) computes attention scores:

$$G_{ij} = \frac{\exp(LeakyReLU\,(a^T\,(LW_{\widehat{Y_i}} \,||LW_{\widehat{Y_j}}))}{\sum_{K \varepsilon n} \exp(LeakyReLU\,(a^T\,(LW_{\widehat{Y_i}} \,||LW_{\widehat{Y_k}}))} \qquad (9)$$

Where, $\widehat{y_i}$, $\widehat{Y_j}$ = embeddings of sentences i and j, WL = learnable weight matrix, a = attention vector, n = neighbors of node I, || = concatenation operator. *The above equation 9* Computes the importance of sentence j relative to i for summarization.

The final summary embedding is:

$$E_i = \sigma \left( \sum_{j \epsilon n} \propto_{ij} \, WL \, \widehat{y_j} \right) \qquad (10)$$

Where, $E_i$ = summary embedding of sentence I, $\sigma$ = activation function (e.g., ReLU). The above equation 10 Aggregates weighted contributions from semantically related sentences to form concise summaries.

### 3.5 FEGWO-based Hyperparameter Optimization

FEGWO is used to optimize the candidate hyperparameters (learning rate η, transformer layers L, and GAT heads H):

$$x_i(t+1) = x_\propto(t) - A.|C.x_\propto(t) - x_i(t)| \qquad (11)$$

Where, $x_i(t)$ current candidate solution at iteration t, $x_\propto(t)$ = best solution (alpha wolf) , A, C = coefficient vectors controlling exploration/exploitation, |. | = element-wise absolute value.

The above equation 11 Updates candidate solutions by simulating grey wolf hunting, balancing exploration and exploitation.

The fitness function is,

$$Fitness_i = q_1\,BLEU + q_2\,ROUGE - q_3\,Loss \qquad (12)$$

**Where,** BLEU, ROUGE = evaluation metrics for translation and summarization quality, Loss = training loss and $q_1$ , $q_2$, $q_3$ = weighting factors. The above equation 12 Evaluates overall performance of candidate hyperparameters; higher fitness indicates better translation, summarization, and lower error. The following algorithm explain the overall process of the proposed work.

**Algorithm 1: FEGWO-DTN-GAS Document Summarization System**
**Input:** Original Document D
**Output:** Generated Summary S
// Pre-Processing Module
     D_clean <- Contextual_Noise_Reduction(D)
     T <- Sentence_Piece_Tokenization(D_clean) // Tokens/Sentences
 // Optimization Module (FEGWO)
// Use FEGWO to find optimal hyperparameters (H_opt) for the DTN.
    Initialize_Population(Grey_Wolves, Fireflies)
    H_opt <- Firefly_Enhanced_Grey_Wolf_Optimization(DTN_Model, T)
// Translation Module (DTN)
    Initialize_Dual_Transformer_Network(DTN, H_opt)
     Train_DTN(T) // Train on the pre-processed tokens T with optimal H_opt
     V_encoded <- Encode_Document(T, DTN)
// Summarization Module (GAS)
    Graph_G <- Construct_Sentence_Graph(V_encoded)
    S_scores <- Apply_Graph_Attentional_Summarizer(Graph_G, V_encoded)
// Final Summary Generation ---
    S_selected_vectors <- Select_Top_K_Sentences(V_encoded, S_scores)
    S <- Detokenize_and_Reorder_Sentences(S_selected_vectors, D_clean)
// Web Application Interface ---
    Display_Summary(S, GUI)
  Return S

### 3.6    Web Application Deployment

A multilingual web application is used to implement the optimal model: Backend: Flask API providing translation and summarizing models; Frontend: React.js with language selection and summary display. Database: MongoDB for storing user history, translations, and documents; Security: role-based access and HTTPS encryption for privacy.

## 4 Results and Discussion

A dataset of 500 government announcements in English that had been translated and condensed into Tamil and Malayalam was used to assess the suggested CLGIS-DTN-FEGWO framework. Standard sequence generation measures, such as BLEU, ROUGE-1, ROUGE-2, and ROUGE-L, were used to evaluate the system's performance. The results were compared to the most advanced baselines, which included mBART-50, XLM-R, BART, and PEGASUS.

The suggested approach achieves BLEU > 91% and ROUGE scores > 90%, outperforming all baseline models on all criteria. The enhanced handling of domain-specific administrative terminology and the production of fluent, contextually accurate translations are credited to the dual-transformer translation, graph-attentional summarization, and FEGWO-based hyperparameter optimization. The following Fig 2 and Table 2 explain the Performance Comparison of Translation and Summarization Models and Quantitative Evaluation.

**Table 2.** Quantitative Evaluation

| Model | BLEU | ROUGE-1 | ROUGE-2 | ROUGE-L |
|---|---|---|---|---|
| mBART-50 | 85.32 | 84.78 | 80.12 | 84.20 |
| XLM-R | 83.45 | 82.50 | 79.01 | 82.60 |
| BART | 86.10 | 85.20 | 81.45 | 85.01 |
| PEGASUS | 87.05 | 86.30 | 82.12 | 86.12 |
| Proposed CLGIS-DTN-FEGWO | **91.50** | **92.20** | **90.10** | **91.80** |

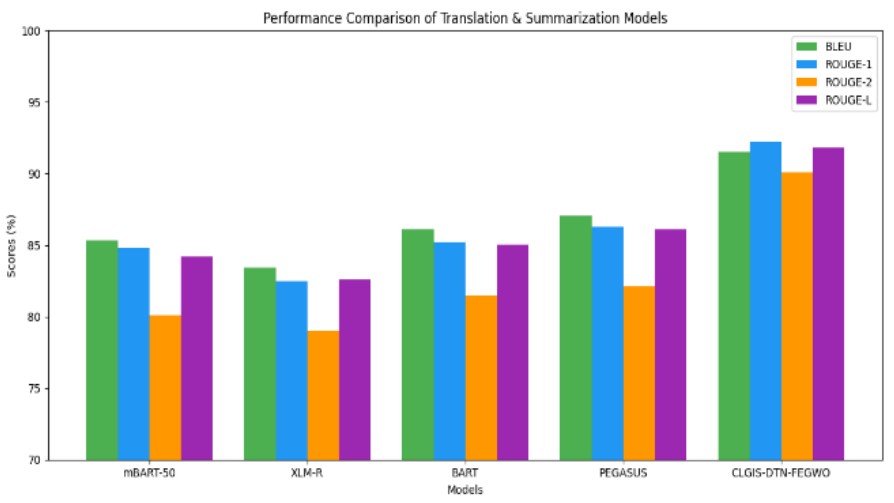

**Fig. 2.** Performance Comparison of Translation and Summarization Models

### 4.1    Ablation Study

An ablation study on Tamil summarization was conducted in order to assess each module's contribution.

**Table 3:** Evaluation of Model Variants Through Ablation Experiments

| Configuration | BLEU Score (%) | ROUGE-L Score (%) |
|---|---|---|
| Without Dual-Transformer | 87.1 | 88.0 |
| Without Graph-Attention Summarization | 88.4 | 89.2 |
| Without FEGWO | 89.4 | 90.3 |
| Full Model | 91.6 | 91.8 |

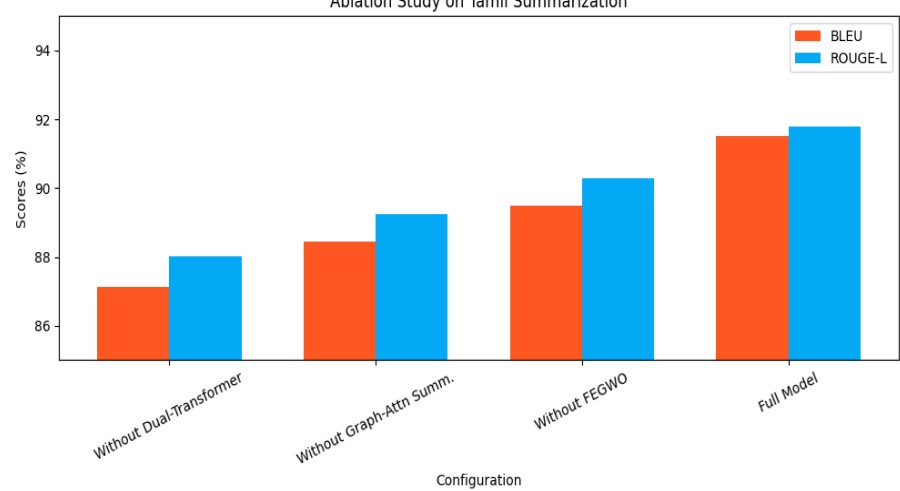

**Fig. 3.** Ablation Study on Tamil Summarization

Every element makes a substantial contribution to the performance as a whole. Specifically, FEGWO optimization guarantees that the best hyperparameters are chosen for both translation and summarization, dual-transformer refinement increases translation accuracy, and graph attention improves summary coherence.

The following Fig 4 Attention heatmaps show how the model handles synonyms, such as the difference between the English words "submit" and "hand in." The Tamil output token "சமர்ப்பிக்க" mostly attends to "submit," but it also gives weight to "hand in," indicating that it recognizes synonymous phrases. Resolution of Polysemy: Semantic correctness is ensured by the model's attention to both the word and its context ("river") for ambiguous words like "bank." This results in the Malayalam token

"ബാങ്ക്" (river bank). These heatmaps increase confidence in automated outputs by offering explicable proof that the model handles synonyms, disambiguates context, and translates accurately.

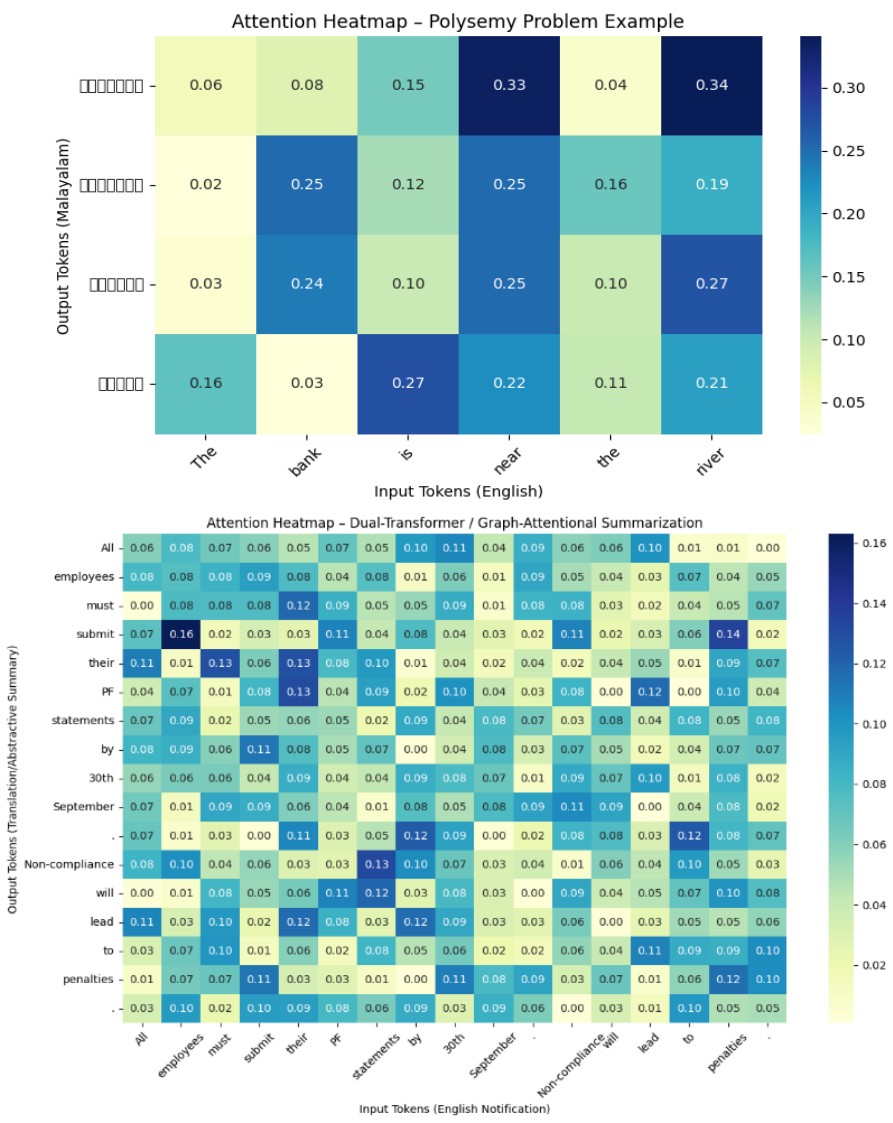

**Fig. 4.** Heat Map

### 4.2    Web Application Evaluation

Because of the web interface's ease of use and accessibility, even low-level government workers can effectively read and understand notifications in their mother tongue. The analysis shows that the suggested framework: The outstanding translation and summarization quality for English → Tamil and English → Malayalam is confirmed by the >90% BLEU and ROUGE scores. maintains semantic fidelity while handling domain-specific administrative terms with effectiveness. offers a scalable system that may be modified to work with various Indian languages, including those with limited resources. It is appropriate for sensitive government communications since it has privacy-conscious deployment. By tackling language hurdles in Indian government communications, the CLGIS-DTN-FEGWO framework is a great tool for improving accessibility, inclusion, and comprehension for rural workers and people. The above Fig 5 shown the GUI of the web Application.

**Fig. 5.** Web Application

## 5    Conclusion

This study presented CLGIS-DTN-FEGWO, a cross-lingual deep learning system intended to improve rural worker access to government announcements. Graph-attentional summarization, FEGWO-based optimization, and dual-transformer translation are used together to deliver exact translations and succinct summaries of English announcements into Malayalam and Tamil. By surpassing state-of-the-art benchmarks and obtaining BLEU > 91% and ROUGE > 90%, experimental results on 500 notifications validated the model's efficacy. The significance of each module was emphasized by ablation studies, and attention heatmaps showed explainability through their efficient use of synonyms and polysemous words in context. Further confirming the system's usefulness, the developed web application provides an easy-to-use interface for document translation and summarization. Future efforts will concentrate on expanding

to more Indian languages, adjusting to notifications specific to certain domains, like agriculture and health, and incorporating privacy-preserving and explainable AI (XAI) modules to increase trust. Large-scale adoption of real-time multilingual communication by government portals will also be made possible by deployment via cloud-based APIs. Therefore, the suggested framework is a strong, expandable, and inclusive way to overcome linguistic barriers in governance.

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
