# OpenReview forum: "Agentic Framework for Multilingual Summarization of Government Notifications Using Firefly-Optimized Dual Transformers"
_SPELLL.org/2025/Workshop/LC4 — LC4 2025 Oral_

### Official Review · Reviewer_QpBn · 2025-11-03
**The paper addresses an important and socially relevant problem of improving access to government notifications for rural and non-English-speaking communities in India.**

**Rating:** 6
**Confidence:** 5

**Review:**

The paper addresses an important and socially relevant problem of improving access to government notifications for rural and non-English-speaking communities in India. The proposed framework demonstrates strong engineering integration by combining contextual preprocessing, a dual-transformer translation module, graph-attentional summarization, and meta-heuristic hyperparameter optimization, and further extends into an explainable and privacy-aware web deployment. Results show consistent improvements over standard multilingual baselines, and the ablation study helps attribute performance gains to each system component. However, several concerns arise regarding the novelty and rigor of the work. Most core components—XLM-R embeddings, translation refinement, Graph Attention Networks, and Firefly/Grey-Wolf optimization—are established methods, making the contribution more about system integration than algorithmic innovation. The evaluation is limited to a relatively small dataset of 500 notifications, and the very high BLEU/ROUGE scores may signal potential overfitting, calling for more rigorous human evaluation, statistical significance analysis, and comparison with stronger Indic baselines such as IndicTrans2 or mT5. The mathematical expressions also require careful revision for consistency and clarity. While the results are promising, the paper would benefit from a clearer articulation of its unique technical contributions, expanded evaluation, and improved writing quality to enhance credibility and impact.

---

### Official Review · Reviewer_rHcJ · 2025-11-03
**Agentic Framework for Multilingual Summarization of Government Notifications Using Firefly-Optimized Dual Transformers**

**Rating:** 7
**Confidence:** 3

**Review:**

- The acronym CLGIS-DTN-FEGWO is too long , if possible, this can be reduced
- some notation usage  is found to be  inconsistent (e.g.  in Equations 9 and 10).
- What would be complexity  and computational cost of this system with  Dual Transformers, GAT, FEGWO ?
- Add  details on the source, diversity,  annotation process of the 500 government notifications used.
- Justify the rationale behind the choice of weights q1, q2 and q3.
- Add a discussion on training data,  fine-tuning details for baselines.
- Have any usability testing done or feedback from actual government employees  taken? - add such feedback to the work.
- For reproducibility,  provide details of code, dataset, hyperparameters.